# Plasma Bile Acid Profiling and Modulation of Secreted Mucin 5AC in Cholangiocarcinoma

**DOI:** 10.3390/ijms241612794

**Published:** 2023-08-14

**Authors:** Elisa Danese, Patricia M.-J. Lievens, Andrea Padoan, Denise Peserico, Roberta Galavotti, Davide Negrini, Matteo Gelati, Simone Conci, Andrea Ruzzenente, Gian Luca Salvagno, Giuseppe Lippi

**Affiliations:** 1Section of Clinical Biochemistry, Department of Engineering for Innovation Medicine, University of Verona, 37129 Verona, Italy; denise.peserico@univr.it (D.P.); davide.negrini@aovr.veneto.it (D.N.); matteo.gelati@univr.it (M.G.); gianluca.salvagno@univr.it (G.L.S.); giuseppe.lippi@univr.it (G.L.); 2Section of Biology and Genetics, Department of Neurosciences, Biomedicine and Movement Sciences, University of Verona, 37129 Verona, Italy; patricia.lievens@univr.it (P.M.-J.L.); roberta.galavotti@univr.it (R.G.); 3Department of Medicine-DIMED, University of Padova, 35122 Padova, Italy; andrea.padoan@unipd.it; 4Section of Hepatobiliary Surgery, Department of Surgery, Dentistry, Pediatrics and Gynecology, University of Verona, 37129 Verona, Italy; simone.conci@univr.it (S.C.); andrea.ruzzenente@univr.it (A.R.)

**Keywords:** bile acids, cholangiocarcinoma, diagnostic biomarkers, MUC5AC

## Abstract

Studies investigating the potential role of circulating bile acids (BAs) as diagnostic biomarkers for cholangiocarcinoma (CCA) are sparse and existing data do not adjust for confounding variables. Furthermore, the mechanism by which BAs affect the expression of the oncogenic mucin 5AC (MUC5AC) has never been investigated. We performed a case–control study to characterise the profile of circulating BAs in patients with CCA (*n* = 68) and benign biliary disease (BBD, *n* = 48) with a validated liquid chromatography–tandem mass spectrometry technique. Odd ratios (OR) for CCA associations were calculated with multivariable logistic regression models based on a directed acyclic graph structure learning algorithm. The most promising BAs were then tested in an in vitro study to investigate their interplay in modulating MUC5AC expression. The total concentration of BAs was markedly higher in patients with CCA compared with BBD controls and accompanied by a shift in BAs profile toward a higher proportion of primary conjugated BAs (OR = 1.50, CI: 1.14 to 1.96, *p* = 0.003), especially taurochenodeoxycholic acid (TCDCA, OR = 42.29, CI: 3.54 to 504.63, *p* = 0.003) after multiple adjustments. Western blot analysis of secreted MUC5AC in human primary cholangiocytes treated with primary conjugated BAs or with TCDCA alone allowed us to identify a novel 230 kDa isoform, possibly representing a post-translationally modified MUC5AC specie.

## 1. Introduction

Cholangiocarcinomas (CCAs) comprise a heterogeneous group of malignancies derived from the epithelial lining of the biliary tree. According to their anatomical location, CCAs are classified as intrahepatic (iCCA), perihilar (pCCA), and distal CCA (dCCA), which have particular similarities but also important inter-tumour and intra-tumor heterogeneity [1].

The incidence of CCA is increasing globally, currently accounting for ~15% of all primary liver cancers and ~3% of gastrointestinal malignancies [2]. Silent presentation and a highly aggressive nature and refractoriness to chemotherapy of these cancers contribute to their increasing mortality worldwide [3].

The pathogenesis of CCA is a complex, multistep process characterised by the dysregulation of various signalling networks. Although many efforts have been made in recent years to better understand the biology of CCA, the intricate network of molecular mechanisms responsible for the early and widespread spread of this malignant tumour has not yet been elucidated [4]. A better understanding of CCA carcinogenesis, tumour-stroma interactions, and key molecular pathways, would enable the development of targeted, individualised therapies and early diagnosis.

Recently, bile acids (BAs) have gained increased interest due to their potential involvement in cancer development and progression [5]. Several studies demonstrated an association between BAs and some forms of liver and gastrointestinal cancer, mostly oesophageal, gastric, and colon cancers [6,7,8,9].

BAs are amphiphilic molecules and are the main component of bile along with cholesterol, phospholipids, and bilirubin [10]. Primary BAs are synthesised in the liver from their precursor molecule cholesterol, conjugated with taurine or glycine, stored in the gall bladder, and then secreted into the bile duct and duodenum. Excreted BAs are largely deconjugated, the majority are reabsorbed via the entero-hepatic circulation and some reach the colon where are converted to secondary BA by gut microbial action before being largely reabsorbed [11].

To date, only a few studies have attempted to characterise the BAs profiles in plasma or serum in CCA and benign biliary disease (BBD) patients, providing discrepant results [12,13,14,15], partly due to a lack of reliable methods for BAs quantification, relatively small sample size of study cohorts, and a lack of correction for confounding variables.

The mechanisms by which BAs exert their carcinogenic effects have not been elucidated, although some evidence has recently emerged in the literature. First, due to their structure, BAs may act as detergents that disrupt the lipid bilayer of cells, potentially allowing carcinogenic substances to enter the cell. Other hypotheses concern the ability of BAs to cause oxidative damage (ROS), epithelial proliferation, cell death, signalling activation, and DNA instability [16]. In addition, BAs have been shown to be involved in tumour progression by inducing changes in oncogenic mucins expression [17,18]. In particular, BAs modulate the expression levels of mucins such as MUC1, MUC2, MUC4, and MUC5AC in oesophageal, gastric, and colon cancers [8,19,20,21,22]. Interestingly, the vast majority of CCA in humans is represented by mucin-producing adenocarcinoma. Mucin 5AC (MUC5AC) is rarely expressed in normal biliary epithelium, but is upregulated in biliary tract cancers [23,24,25] and its enhanced synthesis is associated with unfavourable outcomes [26]. Importantly, MUC5AC is not expressed in hepatocellular carcinoma (HCC), suggesting a different biological pathway and assigning this biomarker an important role in diagnosis differentiation [27].

Moreover, being involved in the process of epithelial-to-mesenchymal transition (EMT), MUC5AC has been demonstrated to be a late marker of disease, with a major role in invasion and metastasis [11,28,29,30,31].

This study aimed to characterise the profile of circulating BAs in patients with CCA and BBD and to investigate the efficacy of plasma BAs as diagnostic biomarkers for CCA. Since the modulation of BAs largely depends on multiple confounders, a directed acyclic graph structure learning algorithm was proposed to evaluate the diagnostic accuracy of circulating BAs. The most promising BAs were then tested in an in vitro study to evaluate their effect on modulating MUC5AC expression in CCA cell lines and normal cholangiocytes.

## 2. Results

### 2.1. Clinical Study

#### 2.1.1. Study Population

Clinical and pathological characteristics of patients with benign biliary diseases (BBD, *n* = 42) and patients with CCA (*n* = 68) are reported in Table 1. Among patients eligible for a surgical approach (*n* = 51), 20 had T1/T2 tumours and 31 T3/T4 tumours by final histopathologic examination. Age was not statistically significantly associated with gender (Student’s *t* = −1.9, *p* = 0.06). The presence of malignant lesions was associated with older age (Student’s *t* = −3.92, *p* < 0.001) but not with gender (Fisher’s exact test, *p* = 0.168).

#### 2.1.2. BAs Profiles in Plasma Samples

A previously validated method in liquid chromatography–tandem mass spectrometry (LC-MS/MS) was used to quantify 15 species of BAs in plasma samples [32]. The total BA concentration was markedly increased in patients with CCA compared with controls (43.1 ± 93.1 vs. 6.2 ± 8.8 µmol/L, *p* < 0.0001). The average concentration of each BA, along with le levels of total bilirubin and total cholesterol are presented in Appendix A. Among the 12 BAs most represented in human plasma, those showing the most significant increase in the CCA group compared with BBD were glycochenodeoxycholic acid (GCDCA), taurochenodeoxycholic acid (TCDCA), glycocholic acid (GCA) and taurocholic acid (TCA) (*p* < 0.0001 for all). The levels of taurodeoxycholic acid (TDCA) and tauroursodeoxycholic acid (TUDCA) were also significantly higher in the CCA group compared with the BBD group (*p* = 0.0257 and *p* = 0.0018 respectively). Conversely, the level of DCA was higher in BBD compared to CCA patients (*p* = 0.0011), while no appreciable differences were found for the other BAs (Figure 1).

After categorising BAs according to the site of synthesis and the state of conjugation, we found a significant difference in the average composition of BAs between the study groups. The proportion of primarily conjugated BAs was significantly higher in CCA patients than in BBD patients, accounting for 75% and 45% of total BAs, respectively (*p* < 0.005). The proportion of primary and secondary unconjugated BAs decreased from 16% and 14%, respectively, in BBD patients, to 1% for both BAs groups in CCA patients. Finally, the proportion of secondary conjugated BAs remained similar in both groups, accounting for 24% of total BAs in BBD patients and 20% in CCA patients (Figure 2 and Table 2).

#### 2.1.3. Understanding of Variables and Confounders

The directed acyclic graph (DAG) technique was used to inspect the relationships between all the studied variables. The construction of DAG was performed by agreement of three operators, DN, ED, and AP, whilst disagreement as well as the final model approval was achieved by a fourth operator (GL). The final DAG which best represents the evaluated model is shown in Figure 3.

In particular, DAG was structured on the basis of the following considerations: firstly, being cholesterol within the pathway of BAs formation, it was considered directly linked to BAs; age and gender were widely described to be associated with cancers, including CCA [33] so that they were considered the most important confounders. BAs were further linked to age and gender, since the concentration of BAs changes during aging [34], and is higher in men than in women [35]. Cholesterol increases with age, especially in males, so it was associated with both age and gender. Finally, bilirubin and BAs are linked to being involved in cholestasis and thus being associated with malignant tumours and age as well.

#### 2.1.4. Statistical Model Results

Table 3 reports the significant association between BAs plasma levels and CCA, adjusted for age, gender, total cholesterol, and total bilirubin according to the DAG model described above and in the method section. Specifically, the odds ratio (OR) reported in Table 3 was derived by different statistical analyses, repeated using the same model changing plasma BA each time (full results in Appendix A). We found a significant increase in the primary conjugated BAs group in the malignant compared to the benign group (*p* = 0.003). Among this group, TCA, TCDCA, and GCA showed a significant increase also when individually considered, while GCDCA was not strictly significant (*p* = 0.084) but moved alongside its physiological group. Other BA groups that were significantly associated with CCA were the following: the ratio of primary to secondary BAs, the ratio of conjugated to non-conjugated BAs, and the ratio of primary conjugated to secondary conjugated BAs, with an increase in the numerator over the denominator of each of the three significant ratios.

### 2.2. In Vitro Study

MUC5AC is a highly glycosylated protein comprising several isoforms ranging in molecular weight from about 50 to 700 kDa As a secreted gel-forming mucin, it is released out of the cell as it has the property to create a dense viscos-elastic mucus barrier, which covers many epithelia and protects from bacteria [36] and other injuries. To exert this activity, MUC5AC undergoes acidic pH-triggered cleavage in the C-terminal GDPH sequence during the passage from the late endoplasmic reticulum (ER) to the secretory pathway [37,38], thus generating, upon secretion, isoforms that could cross-link to the epithelial surface or into the mucus gel. In CCA patients MUC5AC appears upregulated, but the exact mechanisms of MUC5AC contributing to CCA carcinogenesis are currently unknown [26,30,31].

Here, we hypothesised that GCA, GCDCA, TCA, and TCDCA could play a direct role in MUC5AC expression. To assess our hypothesis, we planned to treat human CCA-derived HuH28 cells and primary cholangiocytes with the selected BAs. Both cell types were incubated with 200 µM of each BAs or with a mix of the four BAs at a concentration of 50 µM each, for 16 h in the absence of serum. The cells were then collected together with their media, processed as described in the methods section, and subjected to western blot analyses with an anti-MUC5AC antibody. Cell lysates were used to identify MUC5AC expression profiles present in the two different cell lines, while media were assayed for the presence of secreted MUC5AC isoforms. Results on cell lysates are reported in the Appendix A (Appendix A). Western blot analysis of secreted MUC5AC in untreated HuH28 (Figure 4, left panel) showed a prominent band of 270 kDa that appeared slightly broader in size in samples treated with the different BAs and especially with the mix of four BAs (compare line 1 with a line from 2 to 6) but not significantly increased, such as by quantification (Appendix A). Western blot analysis of secreted MUC5AC in untreated human primary cholangiocytes revealed a much less represented 270 kDa isoform (Figure 4, right panel). Interestingly, exposure of primary cholangiocytes to each single BA resulted in a reduction of the 270 kDa band in favour of an isoform of about 230 kDa (Figure 4, lanes 7–11). Of note, when primary cholangiocytes were treated with TCDCA alone or the mix of four BAs, only the secreted 230 kDa mucin isoform was detected (compare lanes 7 and 12).

## 3. Discussion

Increasing evidence suggests a mechanistic link between the disturbance of BA homeostasis and the development of various types of cancer, so the quantification of BAs in bile or preferably in plasma samples was proposed as a new diagnostic tool in CCA patients. In the present study, we applied a sensitive and accurate method for the simultaneous quantification of 15 major BAs in human plasma and described their changes in patients with CCA and BBD.

Our quantitative analysis showed that the total BAs pool size was markedly increased in CCA compared to BBD patients, a finding consistent with previous data [39,40]. Interestingly, such an increase was almost entirely due to the increase in primary conjugated BAs, whose average concentration was 10-fold higher in CCA patients compared to the BBD group, and to a minor extent to secondary conjugated BAs, whose concentration in CCA patients was 6-fold higher than that found in BBD group. On the other hand, the average concentration of both primary and secondary unconjugated BAs was nearly halved in the CCA group with respect to the BBD group (Table 2). The direct consequence of this modulation was a marked inconsistency in the average proportion of BAs among the two study groups with the BAs pool size of CCA patients resulting almost entirely constituted by primary (75%) and secondary (24%) conjugated BAs (Figure 2). Consistently with our results, Zang and coauthors found that primary conjugated BAs constituted more than 97% of total plasma BAs [13], while Song and coauthors found that primary conjugated BAs accounted for 75% of BAs pool in bile samples from CCA patients [14]. As earlier suggested, the increase in the proportion of primary conjugated BAs in CCA patients compared to healthy or BBD patients could be partially due to a decrease or block of bile excretion in BAs and bilirubin, as CCA patients often present obstructive jaundice [41]. 

To date, only one study revealed the importance of BAs as independent biomarkers of CCA after categorising patients according to total bilirubin levels [40]. In this study, however, the authors evaluated only the total BA concentrations and not the BAs profile. Interpreting how variables interact with each other is necessary for modelling the relationship between BAs and CCAs. To fulfil this aim we created a DAG, which enabled the identification of clear or hidden relationships between BAs, the outcome CCA, the confounders (age, gender, and bilirubin) and cholesterol, the latter involved in BA biosynthesis. As a result, the DAG facilitated the development of correct statistical models by taking into account the physiology of BAs, the presence of cholestasis, and other CCA potential risk factors, including confounders.

By applying this model, we found that the most promising candidate biomarker for differentiating CCA from BBD was TCDCA with an OR of 42.3 (95%CI: 3.54 to 504.63), but also confirmed a predominant role of primary conjugated BAs, taken together and singularly.

Once identified GCA, GCDCA, TCA, and TCDCA as the best biomarkers associated with CCA we investigated their effect on human primary cholangiocytes and HuH28, a well-known CCA cellular model [42,43], with the purpose of assessing changes in MUC5AC expression. Western blot analyses were performed on both cell lysate and media, the latter being used for the analysis of secreted MUC5AC. This experiment revealed a novel MUC5AC isoform of 230 kDa that appeared in media samples from human primary cholangiocytes after treatment with single or mixed BAs, counterbalanced by the 270 kDa isoform that was present in untreated cells. Whereas the 270-kDa band was still weakly detectable in samples treated with GCA, GCDCA, and TCA, treatment with TCDCA alone or the four mixed BAs resulted in unequivocal detection of only the 230-kDa isoform, thus indicating an enhanced effect of TCDCA, the BA which was most strongly associated with CCA in the clinical study. A secreted MUC5AC isoform of 270-kDa was also clearly detected in both treated and untreated HuH28 cell lines. Although different in size, the 230 kDa band observed in treated cholangiocytes and the 270-kDa band observed in both treated and untreated HuH28 cell lines presented a similar broad migration pattern consistent with a relatively high degree of glycosylation. These observations suggest a putative role played by the tested BAs and mostly by TCDCA either in the proteolytic cleavage of MUC5AC or in other MUC5AC processing, yet to be identified, but possibly related to glycosylation [44]. 

Although speculative, it can be hypothesised that the 230 kDa isoform observed in primary cholangiocytes may undergo additional post-translational modifications as the BAs injury progresses, possibly including heavier glycosylation that would slow its migration to an apparent size of approximately 270 kDa, which would explain why this band remains unchanged after treatment with BAs in HuH28 cell line. This hypothesis is supported by the recent observation that MUC5AC consists of numerous glycoforms including less-glycosylated immature isoforms and heavily glycosylated mature isoforms [45]. Further analysis with antibodies specific for both isoforms may be useful to demonstrate the appearance of aberrant glycosylation MUC5AC variants during malignant transformation.

Nevertheless, we cannot exclude the possibility that, despite a similar migration profile, the 270 and 230 kDa bands found in the two different cell types correspond to differently processed MUC5AC isoforms. These considerations may be supported by a recent publication suggesting that MUC5AC efflux is accelerated in the pathological state to the extent that autocatalytic cleavage of MUC5AC at the GDPH site does not occur [38]. Although aware of the fact that this study was performed with an antibody unable to identify all MUC5AC isoforms, our results suggest that selected BAs may play a role in the processing of MUC5AC glycoprotein. If confirmed, such results may have important clinical implications. In fact, although it has been largely demonstrated that alterations of MUC5AC are associated with biliary carcinogenesis and that MUC5AC tissue expression might have predictive value in CCA treatment, the mechanism by which such alterations are induced remains to be explored [45]. Further studies are needed to confirm that the secretion of MUC5AC glycoprotein may be modulated by treatment of CCA cells with BAs and to determine whether induction of MUC5AC by BAs can increase the invasion potential of cells in vitro and their metastatic potential in vivo [46]. A more detailed understanding of the precise mechanisms by which BAs induce MUC5AC alterations could thus facilitate the development of chemopreventive strategies to diminish the risk of carcinogenesis and metastasis.

HuH28 and primary cholangiocytes used in this study represent a first attempt to find suitable models to approach the complex process leading to CCA onset or progression. HuH28 cell line was established as a bile duct cell line derived from a cholagiocellular carcinoma in 1988 [47]. Since then, HuH28 cells have been widely used as a model to study pro-tumourigenic mechanisms related to tyrosine kinase-induced signalling pathways in CCA [48], but also to test tumour growth inhibitors [49]. To our knowledge, no studies reported investigations on secreted MUC5AC in this cell line, so far. Future studies may take advantage of published data on activated signalling pathways in HuH28 to investigate the interplay between effects promoted by selected BAs and specific signalling activation, now also examining their impact on MUC5AC processing and secretion. In this context, cholangiocytes being non-tumoural biliary epithelial cells may help to address early events induced by BAs. The observation that selected BAs may change the profile of secreted MUC5AC in primary biliary epithelial cells, resulting in a prevailing 230 kDa isoform, although yet to be finely characterised in terms of structure, glycosylation or other post-translational modifications, suggests that early events may contribute to the disruption of the mucin-gel layer protecting epithelial cells. Whether the appearance of the MUC5AC 230 kDa isoform may depend on the activation of an abnormal signalling pathway by selected BAs is yet to be explored. Analyses of other available CCA and primary biliary cells will be required to further support our findings.

In conclusion, our study demonstrates for the first time the important role of circulating BAs as independent biomarkers for CCA, with primary conjugated BAs or TCDCA alone showing the most significative results. Moreover, through an exploratory in vitro study, we provided new information on the potential role of primary conjugated BAs in modulating the expression of different MUC5AC isoforms. Further studies are needed to determine whether BA profiling can serve also in risk stratification of subjects who are at higher risk of CCA development and to better characterise their role in promoting aberrant glycosylation profiles of MUC5AC in healthy and malignant cell lines.

## 4. Materials and Methods

### 4.1. Clinical Study

#### 4.1.1. Patients and Sample Collection

A total of 110 patients with benign biliary diseases (BBD, *n* = 42) and CCA (*n* = 68) were enrolled between September 2020 and December 2021 at the Division of General and Hepatobiliary Surgery of the University Hospital of Verona. Patients who had received prior surgery, chemotherapy, radiotherapy, or immunotherapy treatment were excluded from this study. Other criteria of exclusion were the presence of diabetes, liver cirrhosis, acute liver failure, renal failure, as well as multiple organ failure. The CCA cases were histopathologically ascertained after tissue biopsy. TNM staging was performed according to the eighth edition of the American Joint Committee on Cancer Staging Manual. The BBD patients, mainly including choledocholithiasis, were diagnosed through radiographic evidence, ultrasonography, and/or pathologic features. Plasma samples were collected in a heparin tube before surgery and stored at −80 °C until analysis. The study was approved by the Institutional Review Board of the University Hospital of Verona (24113CESC, 16 May 2017). All subjects provided written informed consent according to the institutional guidelines.

#### 4.1.2. Reagents and Chemicals

LC/MS grade methanol, acetonitrile, isopropanol, ammonium formate, and formic acid were purchased from VWR International (Radnor, PA, USA). Ultra-pure water was purified using a Milli-Q system (Merch Millipore, Darmstadt, Germany). Tauroursodeoxycholic acid (TUDCA), taurocholic acid (TCA), glycoursodeoxycholic acid (GUDCA), glycocholic acid (GCA), taurochenodeoxycholic acid (TCDCA), taurodeoxycholic acid (TDCA), cholic acid (CA), ursodeoxycholic acid (UDCA), glycochenodeoxycholic acid (GCDCA), glycodeoxycholic acid (GDCA), taurolithocholic acid (TLCA), chenodeoxycholic acid (CDCA), glycolithocholic acid (GLCA), deoxycholic acid (DCA), hyodeoxycholic acid (HDCA), taurocholic acid-d4 (d4-TCA), glycocholic acid-d4 (d4-GCA), cholic acid-d4 (d4-CA), ursodeoxycholic acid-d4 (d4-UDCA), chenodeoxycholic acid-d4 (d4-CDCA), and deoxycholic acid-d4 (d4-DCA) were purchased from Sigma–Aldrich (St. Louis, MO, USA) or Cayman Chemical (Ann Arbor, MI, USA). Plasma sample preparation was performed as follows. An aliquot of 200 μL of plasma sample was spiked with 780 μL methanol and 20 μL of the internal standard mix in plastic centrifuge tubes. Samples were vortexed for 20 s, centrifuged for 3 min at 25,000× *g*, and then 200 μL of the supernatant was transferred in a 96-well plate for analysis along with 200 μL of water. 

#### 4.1.3. LC-MS/MS Analysis of BAs

Re-suspended BAs and treated plasma samples were injected into a Cortecs T3 column (2.7 μm, 2.1 mm × 30 mm). LC-MS/MS analysis was performed using an integrated system composed of Acquity UPLC I-Class System FTN coupled with a Xevo TQ-S micro MS/MS (Waters Corporation, Milford, MA, USA) detector operating in multiple reaction monitoring and electrospray negative ionisation mode. Method details have been previously described [32].

#### 4.1.4. Statistical Model

Directed acyclic graph (DAG) technique was used to depict biochemical interactions among total bilirubin, total cholesterol, and BAs and their relationships with the outcome and demographic characteristics. The model then included both studied variables and confounders. DAGitty 3.0 browser-based software [50] was used to design DAGs. This analysis has been extensively used in literature for identifying the pathway of causation in statistical modelling [51]. Using the DAG results, multivariable models were used to explore the effective association between predictors and outcomes. The usage of DAG allowed us to generate models explaining the relationship between BAs and CCA, taking into account the physiology of BA (e.g., cholestasis) and the demographic characteristics (age and gender) as confounders. In further analyses, BAs were also evaluated in groups, summing up their concentrations based on their physiological characteristics: unconjugated primary BAs (CA, CDCA), conjugated primary BAs (TCA, GCA, TCDCA, GCDCA), unconjugated secondary BAs (UDCA, HDCA, DCA), and conjugated secondary BAs (TUDCA, GUDCA, TDCA, GDCA, TLCA, GLCA). The ratio between BA concentration and total cholesterol was added to the model to strengthen the importance of total cholesterol involvement in BA synthesis pathways.

#### 4.1.5. Statistical Analysis

Microsoft Excel 365 (Microsoft Corporation, Redmond, WA, USA) was used for data collection. For graphical data exploration and box-blot GraphPad Prism (ver5 and graphs R 4.0.5 64-bit was used (R Foundation for Statistical Computing, Wien, Austria) with Rstudio 1.4.1106 (Rstudio, Boston, MA, USA) and packages “ggplot2” [52] and “readxl” [53]. Statistical inferential analyses were performed using the software Stata MP 16.2 (StataCorp, College Station, TX, USA). Mean, median, standard deviation (SD), and interquartile range (IQR) were used to summarise and describe data. A paired sample t-test or nonparametric Wilcoxon’s signed rank test was used to test the difference between cases and controls. DAG analyses were used to model the relationship between the BAs and outcome using logistic regression, including in the model the following terms: (a) age and gender as confounders, (b) total bilirubin, and the ratio between BAs and total cholesterol. Multivariable analyses were performed for each BA (including the unconjugated primary/secondary BAs and the unconjugated primary/secondary BAs and the derived ratios as specified above), while only the significant models were considered and reported. Odds ratios were used to quantify the association between biomarkers (BAs) and outcome (CCA) 

### 4.2. In Vitro Study

#### 4.2.1. Cell Cultures

Human lung adenocarcinoma cells (A549) were purchased from American Type Culture Collection (ATCC, Manassas, VA, USA) and cultured in DMEM medium (Invitrogen, Carlsbad, CA, USA) supplemented with 10% of Fetal Bovine Serum (FBS), 2 mM L-glutamine and 1% penicillin/streptomycin. Human cholangiocarcinoma cells (HuH28) were purchased from the Japanese Collection of Research Bioresources Cell Bank (JCRB Cell Bank) and maintained in RPMI 1640 medium supplemented with GIBCO GlutaMAX™, 10% FBS and 1% penicillin/streptomycin. Human cholangiocytes primary biliary epithelial cells (CELPROGEN, Torrance, CA, USA) were cultured following recommended manufacturer’s protocols with ready-to-use complete growth medium (CELPROGEN, Torrance, CA, USA), according to the Company. Those cells were obtained from the tissue of donors that underwent standard blood infectious disease screening. All cell cultures were maintained in a 5% CO_2_ incubator at 37 °C. 

#### 4.2.2. BAs Treatment

GCA, GCDCA, TCA, and TCDCA were dissolved in a H_2_O:MeOH solution (50:50 *v*/*v*). For the experimental protocol, HuH28 and cholangiocytes were cultured for 16 h in a medium without serum and treated with 200 µM of a single BA or with a mix composed of 50 µM of each selected BA.

#### 4.2.3. Cell Lysates and Media

Culture media were collected and cell debris were removed by centrifugation at 1000× *g* for 6 min, then supernatants were stored at −80 °C. Cells were washed twice with PBS, then scraped in RIPA lysis buffer with protease inhibitor cocktail, and proteins were extracted upon incubation on ice for 30 min. Lysates were centrifugated at 21,000× *g* for 10 min at 4 °C to precipitate insoluble cell debris and the collected supernatants were stored at −80 °C. The protein concentration of lysates was determined using Bicinchoninic Acid Kit for Protein Determination (BCA1-1KT, Sigma-Aldrich, St. Louis, MO, USA). Cell cultured media (300 μL) were lyophilised and resuspended in 30 µL of Sample Buffer 1× (NuPAGE™ LDS Sample Buffer 4×, NP0007, Invitrogen).

#### 4.2.4. Western Blot Analysis

Western blot analyses were performed using XCellSureLock™ Mini-Cell and XCell II™ Blot Module (Invitrogen) devices. Lysates and media were loaded onto NuPAGE™ 3–8% Tris-Acetate gels (Invitrogen) and processed according to the manufacturer’s protocol except for elongating protein transfer time. HiMark™ Pre-Stained Protein Standard (LC5699, Invitrogen) or Precision Plus Protein Dual Color Standards (1610374, Bio-Rad Laboratories, Hercules, CA, USA) was used to assess the protein separation and electrophoresed proteins were transferred onto a polyvinylidene difluoride (PVDF) Transfer Membrane (0.45 μm) (88518, Invitrogen). Media blots were stained with Ponceau S Staining Solution (Thermo Fisher™, Scientific, Waltham, MA, USA) for loading evaluation. Then, blots were blocked in 5% non-fat dried milk (A0830, ITW Reagents, AppliChem GmbH, Darmstadt, Germany) prepared in Tris-buffered saline with 1% Tween 20 (TBS-T) for 1 h at room temperature (RT), then incubated ON at 4 °C with primary antibodies diluted in TBS-T with 3% bovine serum albumin (A6588, ITW Reagents, AppliChem GmbH, Darmstadt, Germany). We used the following antibodies: rabbit anti-MUC5AC (M00612-1, Boster Biological Technology, Pleasanton, CA, USA), mouse anti-β-actin (sc-47778, Santa Cruz Biotechnology, Santa Cruz, CA, USA), secondary anti-Rabbit IgG HRP-conjugated antibody (NA934V, GE Healthcare, Chicago, IL, USA) or anti-Mouse IgG HRP-conjugated antibody (NA931V, HealthCare Technologies, Chicago, Illinois, USAGE HealthCare). Blots were incubated with SuperSignal™ WestPico PLUS Chemiluminescent Substrate (34580, Thermo Fisher Scientific, Waltham, MA, USA) and visualised using Alliance Q9 Advanced (UVITEC Cambridge, UK).

## Figures and Tables

**Figure 1 ijms-24-12794-f001:**
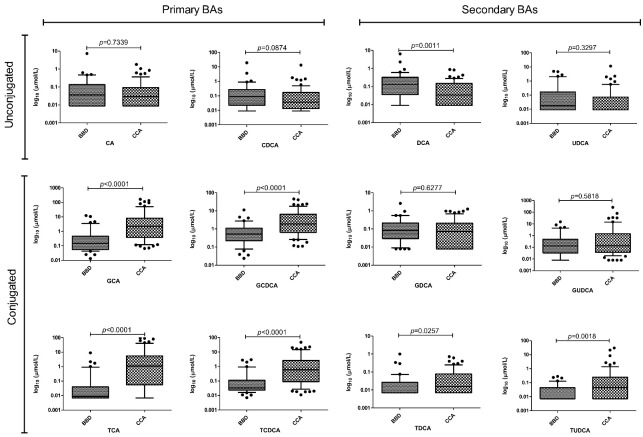
Different changes of plasma BAs in CCA and BBD groups. BAs with values below the limit of quantification of the method in more than 70% of the samples (taurolithocholic acid, TLCA, glycolithocholic acid, GLCA, hyodeoxycholic acid, HDCA) were excluded from the analysis. Data are shown as median and 10th and 90th percentiles. CA: cholic acid, CDCA: chenodeoxycholic acid, UDCA: ursodeoxycholic acid, GUDCA glycoursodeoxycholic acid.

**Figure 2 ijms-24-12794-f002:**
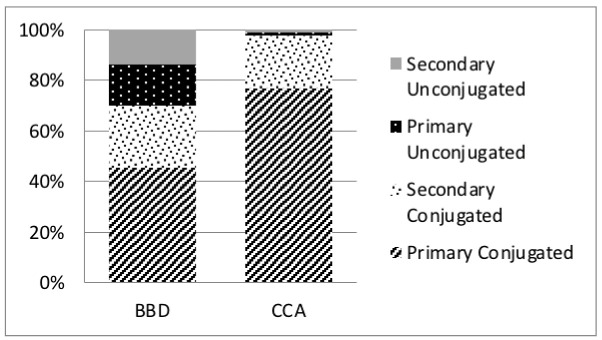
Composition ratio of BAs grouped according to the state of conjugation in BBD and CCA samples.

**Figure 3 ijms-24-12794-f003:**
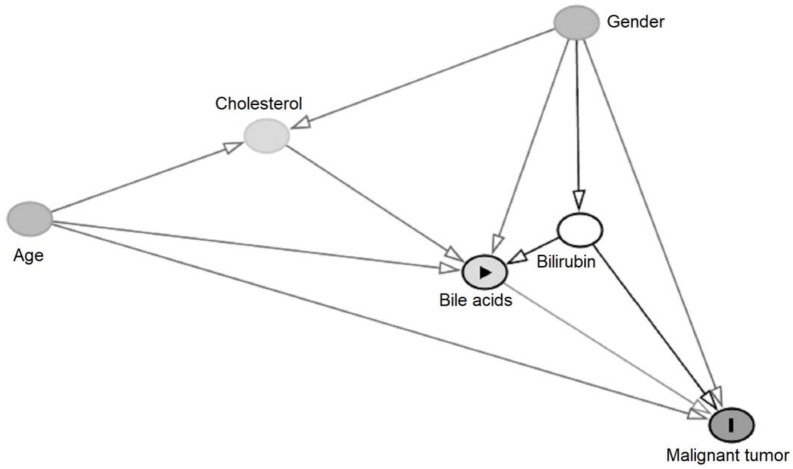
Directed acyclic graph (DAG) showing causal relationships between study variables. Directed Acyclic Graph (DAG), schematising the relationship between BAs (risk factor), CCA (outcome), and the studied variables. Cholesterol is directly linked to BAs since it is involved in the biosynthesis of BAs. Gender, bilirubin, and age were considered as possible confounders, being associated both with BAs and with CCA.

**Figure 4 ijms-24-12794-f004:**
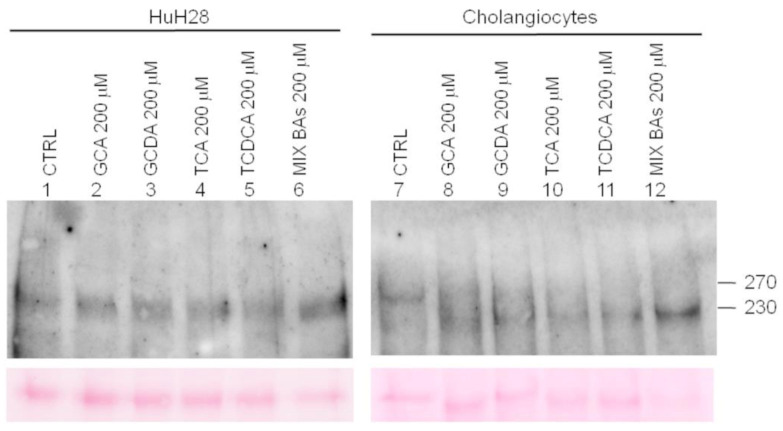
MUC5AC expression profile in HuH28 and cholangiocytes media before and after incubation with selected BAs. HuH28 and human primary cholangiocytes were treated with single and mixed BAs and probed with anti-MUC5AC antibodies. Lower panels: bands detected by Ponceau solution on the membranes, as a loading control. The blots shown here are representative of three independent experiments.

**Table 1 ijms-24-12794-t001:** Demographics and clinicopathological characteristics of patients.

**N. Patients**	110
**Age**	
≤65	48 (43.6%)
>65	62 (56.4%)
**Gender**	
Male	64 (58.2%)
Female	46 (41.8%)
**Type of disease**	
Malignant (CCA)	68 (61.8%)
Benign (BBD)	42 (38.2%)
**Type of benign disease**	
Choledocholithiasis	40 (93%)
Other benign biliary disease	2 (7%)
**Type of malignant disease**	
Perihilar cholangiocarcinoma	38 (55.9%)
Intrahepatic cholangiocarcinoma	15 (22.0%)
Distal Cholangiocarcinoma cancer	15 (22.1%)

**Table 2 ijms-24-12794-t002:** Different concentrations of grouped BAs in CCA and BBD patients.

	Primary Unconjugated (µmol/L)	Primary Conjugated (µmol/L)	Secondary Unconjugated (µmol/L)	Secondary Conjugated (µmol/L)
CCA	0.49	33.57	0.45	9.23
BBD	1.00	2.80	0.86	1.50

**Table 3 ijms-24-12794-t003:** Significant associations between individual or grouped BAs and CCA obtained from models derived from the DAG.

Plasma BAs	OR	95% C.I.	*p*-Value
GCDCA (µmol/L)	2.01	0.91 to 4.42	0.083
TCA (µmol/L)	8.74	1.56 to 49.02	0.014
GCA (µmol/L)	2.56	1.31 to 4.99	0.006
TCDCA (µmol/L)	42.29	3.54 to 504.63	0.003
Primary conjugated BAs (µmol/L)	1.50	1.14 to 1.96	0.003
Primary/Secondary (ratio)	1.14	1.04 to 1.25	0.007
Total Conjugated/Total Unconjugated (ratio)	1.06	1.01 to 1.13	0.046
Primary conjugated/Secondary conjugated (ratio)	1.12	1.03 to 1.21	0.006

## Data Availability

The data presented in this study are available in the Appendix A.

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
