# Peer review of "Plasma Bile Acid Profiling and Modulation of Secreted Mucin 5AC in Cholangiocarcinoma"

_ijms, 2023, doi:10.3390/ijms241612794_

Round 1

Reviewer 1 Report

The authors present a clinical and in vitro study aiming to assess the potential role of circulating BAs as diagnostic biomarkers for CCA. Overall, the manuscript investigates an innovative topic; however, there are several issues I would like to highlight.

Major issues:

-          The Methods section should not be placed at the end of the manuscript. A clear PICOS (Population, Intervention, Comparison, Outcomes, and Study) of the study should be provided with endpoints clearly specified in the methods section.

Methods:

-          The number of included patients should be reported in the results section.

-          Clear endpoints (and outcomes assessed) for the clinical and in vitro analysis should be added.

-          The methodology behind the adjusted analysis should be specified.

Results:

-          A table summarizing demographic information of the study and control groups should be added.

-          Student's t and Fisher's test values could be omitted.

-          A summary of the BAs assessed could be helpful for the reader.

-          Prefer the term "multivariable" to "multivariate." Additionally, multivariable models should be clearly detailed in the manuscript.

Discussion/Conclusions:

-          - What are the clinical implications of your findings? Please comment on that. 

Minor revision needed

Reviewer 2 Report

In this study, Danese et al. seek the association of bile acids with CCA. The association between bile acids and CCA was suggested in previous studies, so the topic is not novel. In addition, there are some issues in this manuscript.

·       This study shows that bile acids are elevated and increase odds ratio in CCA. Data would be stronger if the authors show the correlation between bile acids and pathophysiology of CCA (e.g., correlation between GCA levels with ALT levels or MUC5AC levels).

·       Data would be stronger if the authors perform ROC analysis show AUC scores to show the diagnostic potentials of bile acids.

·       Data would also be stronger if the authors show survival rates of bile acids for CCA patients.

·       The title says as if MUC5AC is important for CCA, but in this study, only in vitro data are shown in Figure 4, and only Figure 4 does not prove anything. If the authors want to include data of MUC5AC, more experimental data should be provided. For example, MUC5AC expression is elevated in CCA tumors in this cohort? Is there the correlation between bile acid levels and MUC5AC expression? Low bile acids and low MUC5AC expression lead to better surivival?

Minor

Reviewer 3 Report

Your study is interesting and provides new insights into the role of bile acids in cholangiocarcinoma. However, I have some major concerns that need to be addressed. Please see below for my comments and suggestions.

- The main finding of your study is bile acid concentrations in cholangiocarnoma and benign biliary diseases.

But the as additional finding, 
GCA and GCDCA can induce MUC5AC expression and secretion in cholangiocarcinoma cells and normal cholangiocytes. However, you did not explain the biological significance and implications of this finding. How does MUC5AC contribute to the development and progression of cholangiocarcinoma? What are the potential clinical applications of modulating MUC5AC expression by bile acids? Please provide more discussion and references on these aspects.

- You showed that GCA and GCDCA can activate the NF-κB pathway, which is involved in the regulation of MUC5AC gene transcription. However, you did not investigate the molecular mechanisms by which bile acids activate NF-κB. How do bile acids interact with the receptors and signaling molecules that mediate NF-κB activation? Are there any other pathways or factors that are involved in bile acid-induced MUC5AC expression? Please perform additional experiments or provide more evidence to support your hypothesis.

- You used two cell lines, HuH28 and primary cholangiocytes, to study the effect of bile acids on MUC5AC expression. However, you did not provide enough information about these cell lines. How were they obtained, characterized, and validated? How representative are they of the human cholangiocarcinoma and normal biliary epithelium? Please provide more details and references on these cell lines.

- You measured the MUC5AC expression by western blot analysis¹[1]. However, you did not quantify the band intensity or normalize it to a loading control. How did you ensure the accuracy and reproducibility of your results? Please provide the quantification and normalization data for your western blot analysis.

- You reported a novel 230-kDa isoform of MUC5AC that appeared in primary cholangiocytes after treatment with bile acids. However, you did not characterize this isoform further. How does it differ from the 270-kDa isoform in terms of structure, function, and glycosylation? What is the significance of this isoform in cholangiocarcinoma? Please perform additional analyses or provide more explanation on this isoform.

Round 2

Reviewer 1 Report

Thank you for submitting an updated version of the manuscript. I have read again your manuscript with great interest. Overall, I think the quality has been improved following reviewers' suggestions. 

Ok

Author Response

We thank the reviewer for the positive response

Reviewer 2 Report

No further comments

none

Author Response

(The authors gave the same response as above.)

Reviewer 3 Report

I have carefully reviewed your changes and responses to the comments of the reviewers and the editor.

I am pleased to inform you that I am satisfied with your revisions.

You have successfully addressed the main issues raised by the reviewers and the editor, and you have improved the clarity, accuracy, and originality of your manuscript. In particular, I appreciate that you have:

- Discussed the limitations and implications of your findings in the context of the current literature on bile acids, MUC5AC, and cholangiocarcinoma. You also cited more relevant references to support your arguments and claims. This enhanced the originality and significance of your paper.

- Corrected some grammatical and spelling errors, and improved the readability and structure of the paper.

I also commend you for providing a detailed response letter to the editor and reviewers, where you thanked them for their time and comments, explained how you addressed each point raised, and provided a scientific rebuttal to some points you disagreed with.

Author Response

We thank the reviewer for the positive response and for  and for appreciating our efforts to improve the manuscript.